# Circulating miRNAs Act as Diagnostic Biomarkers for Bladder Cancer in Urine

**DOI:** 10.3390/ijms22084278

**Published:** 2021-04-20

**Authors:** Jen-Tai Lin, Kuo-Wang Tsai

**Affiliations:** 1Division of Urology, Department of Surgery, Kaohsiung Veterans General Hospital, Kaohsiung 813414, Taiwan; jtlin@vghks.gov.tw; 2Department of Research, Taipei Tzu Chi Hospital, Buddhist Tzu Chi Medical Foundation, New Taipei 23142, Taiwan

**Keywords:** bladder cancer, microRNA, circulating miRNA, biomarker

## Abstract

MicroRNAs (miRNAs) can be secreted into body fluids and have thus been reported as a new type of cancer biomarker. This study aimed to determine whether urinary miRNAs act as noninvasive biomarkers for diagnosing bladder cancer. Small RNA profiles from urine were generated for 10 patients with bladder cancer and 10 healthy controls by using next-generation sequencing. We identified 50 urinary miRNAs that were differentially expressed in bladder cancer compared with controls, comprising 44 upregulated and six downregulated miRNAs. Pathway enrichment analysis revealed that the biological role of these differentially expressed miRNAs might be involved in cancer-associated signaling pathways. Further analysis of the public database revealed that *let-7b-5p*, *miR-149-5p*, *miR-146a-5p*, *miR-193a-5p*, and *miR-423-5p* were significantly increased in bladder cancer compared with corresponding adjacent normal tissues. Furthermore, high *miR-149-5p* and *miR-193a-5p* expression was significantly correlated with poor overall survival in patients with bladder cancer. The qRT-PCR approach revealed that the expression levels of *let-7b-5p*, *miR-149-5p*, *miR-146a-5p* and *miR-423-5p* were significantly increased in the urine of patients with bladder cancer compared with those of controls. Although our results indicated that urinary miRNAs are promising biomarkers for diagnosing bladder cancer, this must be validated in larger cohorts in the future.

## 1. Introduction

Among men worldwide, bladder cancer ranks as the ninth most common cancer and 11th in terms of mortality [1]. The male-to-female bladder cancer incidence ratio is approximately 5:2, with smoking considered the reason for male predominance [2]. Most patients with advanced bladder cancer present with painless gross hematuria sometimes accompanied by the sensation of urgently needing to urinate, and even urge incontinence. In terms of survival outcomes, the five-year relative survival rates of advanced bladder cancer have seen little improvement. When a patient is suspected of having bladder cancer, the initial assessment involves cystoscopy and imaging of the urinary tract through techniques such as sonography and computed tomography urography. However, tiny papillary tumors and carcinoma in situ are difficult to detect using standard white-light cystoscopy or urinary tract imaging, which may account for the misdiagnosis or early recurrence of bladder cancer. This difficulty has led scientists to develop new cystoscopic techniques, such as narrow-band imaging cystoscopy and photodynamic diagnosis. Previous studies have identified numerous noninvasive biomarkers in urine that can serve as diagnostic indicators of bladder cancer [3]. Comprehensive screening approaches have been applied to urine samples to identify cell-free DNA, cell-free RNA, small RNA, or DNA methylation in cells, including DNA methylation of PCDH17 and POU4F2 [4], as well as determine circulating mRNA levels of IGFBP5, HOXA13, MDK, CDK1, and CXCR2 [5] and the mutation status of FGFR3 and TERT promoters [6]. Some urine-related biomarkers have been approved for clinical use by the United States Food and Drug Administration (FDA), such as nuclear matrix protein 22 [7], human complement factor H-related protein [8], carcinoembryonic antigen [9], and sulfated mucin glycoprotein [10]. These biomarkers are typically used in conjunction with cystoscopy to increase the diagnostic sensitivity and specificity for bladder cancer. However, these FDA-approved biomarkers have the shortcoming of a high false-positive rate in patients with inflammatory conditions. To date, the diagnosis of early bladder cancer remains difficult; therefore, the development of noninvasive biomarkers for early diagnosis will be highly beneficial for patients with bladder cancer.

Numerous noncoding RNAs have been constitutively transcribed from human genomic DNA. Among them, small RNAs (approximately 21 to 23 nucleotides long) are microRNAs (miRNAs), which play critical roles in human disease processes [11]. In general, miRNAs exert their biological functions through binding the 3′-untranslated region of protein-coding genes, resulting in the degradation of messenger RNAs or destruction of the protein translation process. Therefore, miRNAs might play distinct roles depending on their targeting genes during cancer progression [12,13,14,15,16,17,18,19,20]. Recently, numerous studies have reported that miRNAs can be secreted from cells through exosome particles into body fluids, including blood, urine, tears, and gastric juice [21,22,23]. As circulating miRNAs are packaged in exosome particles, they are protected from RNase degradation. These stable circulating miRNAs could act as noninvasive biomarkers reflecting certain physiological statuses, including cancer [22,24]. To date, numerous circulating miRNAs have been detected in urine, and they have potential as noninvasive diagnosis biomarkers for bladder cancer [25,26]. Various miRNAs have been detected in the urine of patients with bladder cancer. Pardini et al. reported that three circulating miRNAs (*let-7c-5p*, *miR-30a-5p*, and *miR-486-5p*) were differentially expressed in all bladder cancer subtypes and were accurate biomarkers for discriminating cases and controls [25]. They concluded that these circulating urinary miRNAs could act as accurate noninvasive biomarkers for the early diagnosis of bladder cancer. Another study established diagnostic panels (*miR-31-5p* and *miR-93-5p*) that enabled the sensitive detection of bladder cancer with areas under the receiver operating characteristic curve (AUCs) of 0.84 and 0.81 in the training and validation cohorts, respectively [27]. In addition, Du et al. analyzed small RNA profiles of urine from patients with bladder cancer and identified a group of miRNAs that were differentially expressed, which included *miR-7-5p*, *miR-22-3p, miR-29a-3p, miR-126-5p, miR-200a-3p, miR-375,* and *miR-423-5p*. An examination of the expression levels of these miRNAs in urine enabled accurate diagnosis of bladder cancer with AUC values of 0.923 and 0.916 in training and validation sets, respectively [28]. According to the aforementioned results, the distribution of urinary miRNAs might differ in bladder cancer in people of different ethnicities. In this study, we sought to determine whether urinary miRNAs can act as noninvasive biomarkers for diagnosing bladder cancer in Taiwanese patients.

## 2. Results

### 2.1. MicroRNA Profiles Were Generated by Next-Generation Sequencing

We collected 180 urine samples, including samples from 100 healthy volunteer controls and 80 patients with bladder cancer. Furthermore, to identify whether circulating miRNAs act as diagnostic biomarkers for bladder cancer, we analyzed small RNA profiles of urine from 10 healthy controls and 10 patients with bladder cancer through next-generation sequencing. A total of 50 circulating urinary miRNAs with differential expression were present in the bladder cancer group compared with the healthy control group (Figure 1A,B). We further identified the putative targets of these differentially expressed miRNAs and subjected them to pathway enrichment analysis. As illustrated in Figure 1C, the biological functions of these miRNA candidates were significantly involved in cancer-related signaling pathways, including those of proteoglycans in cancer, MAPK, TGF-beta, FoxO, colorectal cancer, cellular senescence, the adherens junction, PI3K-Akt, Hippo, autophagy, focal adhesion, and ErbB. Some of these signaling pathways have been reported to have a biological function in bladder cancer progression [29,30].

### 2.2. miRNAs Were Upregulated in Bladder Cancer

Studies have revealed that cancer-associated circulating microRNAs in serum or urine are released through exosomes, which are produced by normal cells and cancerous lesions [11,31]. Therefore, we selected five upregulated circulating miRNAs, namely *let-7b-5p*, miR-146-5p, *miR-149-5p*, *miR-423-5p* and *miR-193a-5p*, for further examination in bladder cancer. As depicted in Figure 2, the expression levels of *let-7b-5p* (*p* value = 0.001), *miR-146a-5p* (*p* value = 0.001), *miR-149-5p* (*p* value < 0.001), *miR-193a-5p* (*p* value = 0.02), and *miR-423-5p* (*p* value < 0.001) were significantly increased in bladder cancer compared with corresponding adjacent normal tissues. Analysis of The Cancer Genomics Atlas (TCGA) database revealed that high *miR-149-5p* (crude hazard ratio [CHR], 1.52; 95% confidence interval [CI] 1.00–2.33; *p* = 0.05) and *miR-193a-5p* expression levels are significantly associated with a poor overall survival of patients with bladder cancer (CHR, 1.90; 95% CI 1.29–2.78; *p* = 0.001; Table 1 and Figure 3). Multivariate logistic analysis revealed that high *miR-149-5p* (adjusted HR [AHR], 1.70; 95% CI, 1.11–2.61; *p* = 0.015) and *miR-193a-5p* (AHR, 1.86; 95% CI, 1.26–2.74; *p* = 0.002) expression levels were significantly associated with a poor survival curve in overall survival (Table 1). Together, *miR-149-5p* and *miR-193a-5p* could act as prognostic biomarkers for bladder cancer.

### 2.3. Four Circulating miRNAs Were Highly Expressed in the Urine of Patients with Bladder Cancer

We further examined whether *let-7b-5p*, *miR-146a-5p*, *miR-149-5p*, *miR-423-5p*, and *miR-193a-5p* could be expressed from bladder cancer cells at excessive levels and detected in urine to investigate whether they could be biomarkers for detecting bladder cancer. We analyzed the concentrations of the five miRNA candidates in 90 healthy controls and 70 patients with bladder cancer. The expression levels were assessed using TaqMan real-time PCR assays. As illustrated in Figure 4, *let-7b-5p* (*p* value = 0.0002), *miR-146a-5p* (*p* value = 0.002), *miR-149-5p* (*p* value = 0.002), and *miR-423-5p* (*p* value = 0.001) were significantly increased in the urine of patients with bladder cancer compared with those of healthy controls. These findings indicate the aberrant expression of *let-7b-5p*, *miR-146a-5p*, *miR-149-5p*, and *miR-423-5p* in bladder cancer, and their circulating abundance indicates that they are potential noninvasive biomarkers for detecting bladder cancer.

## 3. Discussion

Studies have revealed that different expression patterns of circulating miRNAs in body fluids might originate from different cell types under certain physiological statuses [22,24]. Therefore, miRNAs might be useful noninvasive biomarkers for diagnosing human cancers and detecting their recurrence. Our previous studies have reported that circulating miRNAs in serum could act as diagnostic biomarkers for human diseases, including gastric cancer recurrence, bipolar II disorder, and Kawasaki disease [32,33]. Other studies have revealed that several circulating miRNAs could be released into urine and thus could potentially be used as biomarkers for diagnosing human cancers, especially those of the urinary tract [34,35]. Moreover, one study used real-time PCR to measure the levels of miR-145 and miR-200a in the urine of patients with bladder cancer and healthy controls; the authors revealed that these miRNAs exhibited promise as noninvasive biomarkers for the diagnosis of bladder cancer and detection of its recurrence [36]. By using TaqMan Human MicroRNA Arrays, Mengual et al. were the first to globally profile urinary miRNA expression, and they identified a list of dysregulated miRNA candidates for noninvasive diagnostics biomarkers of bladder cancer, including *miR-25*, *miR-18a**, *miR-187*, *miR-204*, *miR142-3p*, and *miR-140-5p* [37]. Ghorbanmehr et al. identified three urinary microRNAs, namely *miR-21-5p*, *miR-141-3p*, and *miR-205-5p*, as potentially strong choices for noninvasive diagnostic biomarkers for bladder and prostate cancer diagnosis [38]. According to the aforementioned findings, different research groups have identified miRNA candidates in urine that are not identical, which may be caused by different ethnicities of participants.

In another study, high levels of miRNAs expressed from tumor lesions were reported to be released into body fluid through extracellular vesicles [39]. Based on this, examining the expression levels of miRNA candidates in bladder cancer tissues and urine samples in the same cohort would have been of great interest. In the present study, we collected only urine samples from patients with bladder cancer; therefore, we did not have data on miRNA expression from bladder cancer tissue from the same patients. Furthermore, we examined the expression levels and clinical impacts of miRNA candidates in bladder cancer by analyzing the TCGA database. Due to this limitation, we were unable to conclude whether the high expression of circulating miRNAs in urine was from bladder cancer lesions.

In the present study, we compared the abundance of miRNAs in urine between healthy controls and patients with bladder cancer, revealing 50 circulating urinary miRNAs with differential expression in patients with bladder cancer. Pathway enrichment analysis indicated that these differentially expressed miRNA candidates were involved in cancer-associated signaling pathways, including the MAPK, PI3K-Akt, focal adhesion, and ErbB signaling pathways. The importance of MAPK, PI3K-Akt, and ErbB signaling in bladder cancer development was previously demonstrated in clinical, mouse, and in vitro cell models [40]. Furthermore, we examined the expression levels of *let-7b-5p*, *miR-149-5p*, *miR-146a-5p*, *miR-193a-5p*, and *miR-423-5p* and found that they were significantly increased in bladder cancer compared with adjacent normal tissues. Studies have reported that circulating *let-7b-5p* expression exists in human plasma or serum, and thus, *let-7b-5p* is a promising biomarker for detecting cancers, including breast cancer and nasopharyngeal carcinoma [41,42,43,44]. In addition, both circulating *let-7b-5p* and *miR-423-5p* expression in plasma were significantly altered in young women who participated in long-term exercise [45]. Furthermore, recent studies have indicated that *miR-149-5p* has a tumor-suppressive role in the regulation of cancer cell growth in human cancer through modulating GIT1, circNRIP1, and HNF1A-AS [46,47,48].

The present study is the first to report that *miR-149-5p* is significantly increased in bladder cancer and that high *miR-149-5p* expression is strongly associated with poor survival in bladder cancer. Furthermore, we provided novel results that demonstrate *miR-149-5p* in urine as a potential noninvasive biomarker for bladder cancer detection; miRNA-423-5p might play an opposite role in various human cancers; and *miR-423-5p* expression could promote gastric cancer cell growth and invasion ability by silencing TFF1 expression [41]. Li et al. reported that *miR-423-5p* was significantly overexpressed and that *miR-423-5p* expression accelerated glioma cell growth, angiogenesis, and invasion by activating AKT and ERK1/2 signaling [49]. By contrast, *miR-423-5p* had significantly low expression in osteosarcoma tissues, and *miR-423-5p* expression could suppress osteosarcoma cell proliferation, colony formation, and invasion ability by targeting STMN1 expression [50]. Tang et al. indicated that expression levels of *miR-423-5p* were significantly decreased in ovarian cancer and the plasma of patients with ovarian cancer compared with those in healthy individuals. Therefore, *miR-423-5p* expression could act as a diagnostic biomarker and have a biological function as a tumor suppressor, inhibiting ovarian cancer cell growth and invasion [51]. To date, the biological function of *miR-423-5p* remains unclear in bladder cancer. In the present study, after analyzing the TCGA database, we reported that *miR-423-5p* expression was significantly increased in bladder cancer tissues. Furthermore, the expression levels of *miR-423-5p* were significantly increased in urine from patients with bladder cancer compared with healthy controls. Our data imply that *miR-423-5p* has an oncogenic role and that urinary *miR-423-5p* expression can act as a diagnostic indicator of bladder cancer. However, these findings remain unclear and should be validated in larger cohorts in the future.

## 4. Materials and Methods

### 4.1. Urine Samples and Small RNA Extraction

Eighty preoperative urine samples were obtained from patients who had undergone surgery in the Division of Urology, Department of Surgery, Kaohsiung Veterans General Hospital. Such clinical samples are routinely collected by the hospital’s Biobank. In addition, 100 urine samples were obtained from healthy volunteer controls recruited in the Taiwan Biobank cohort study. Our study protocol was independently reviewed and approved by the hospital’s institutional review board (IRB number: VGHKS18-CT12-05).

### 4.2. Small RNA Extraction from Urine

Urinary small RNA was extracted from 300 μL of urine using a miRNeasy Serum/Plasma Kit (Qiagen, Valencia, CA, USA) according to the manufacturer’s instructions. RNA was resuspended in 14 μL of RNase-free H_2_O.

### 4.3. Generation of Small RNA Profiles Using Next-Generation Sequencing

A total of 20 urine samples were collected from 10 patients with bladder cancer and 10 healthy controls; 300 μL of each urine sample was then subjected to RNA extraction by using the miRNeasy Serum/Plasma Kit (Qiagen, Valencia, CA, USA). Finally, an Illumina small RNA preparation kit (Illumina, San Diego, CA, USA) was used to prepare a small RNA library, and the Illumina HiSeq platform (Illumina, San Diego, CA, USA) was then used for sequencing. First, quality control was performed on the generated sequence readings to remove low-quality readings. As mentioned, the sequence reads were then modified with 3′ linkers to produce clean reads. To study the miRNA expression profiles in 20 libraries, we mapped the qualified sequence readings back to human pre-miRNA (miRBase 19). We evaluated miRNA expression levels and presented them in transcripts per million (TPM); detailed information has been described in our previous studies [12,52].

### 4.4. Pathway Enrichment Analysis

The putative targets of the differentially expressed circulating miRNAs were identified using a target prediction tool. Then, the targets were subjected to pathway enrichment analysis by using the MIENTURNET (http://userver.bio.uniroma1.it/apps/mienturnet/, accessed on 12 April 2021) [53].

### 4.5. TaqMan Real-Time PCR Assay

After RNA extraction, 2 μL of total RNA was subsequently used in a TaqMan Advanced cDNA Synthesis Kit (Applied Biosystems, Foster City, CA, USA) according to the manufacturer’s instructions. Details on this process were described in our previous study [32].

### 4.6. Statistical Analysis

In this study, the expression levels of miRNA candidates in patients with bladder cancer and healthy controls determined through real-time PCR were evaluated using Student’s *t*-tests. Differences were considered significant when *p* < 0.05. The overall survival curves were examined using the Kaplan–Meier method or a log-rank test. A Cox proportional hazards model was used to determine the independent predictors of survival with factors determined to be significant in a univariate analysis employed as covariates.

## Figures and Tables

**Figure 1 ijms-22-04278-f001:**
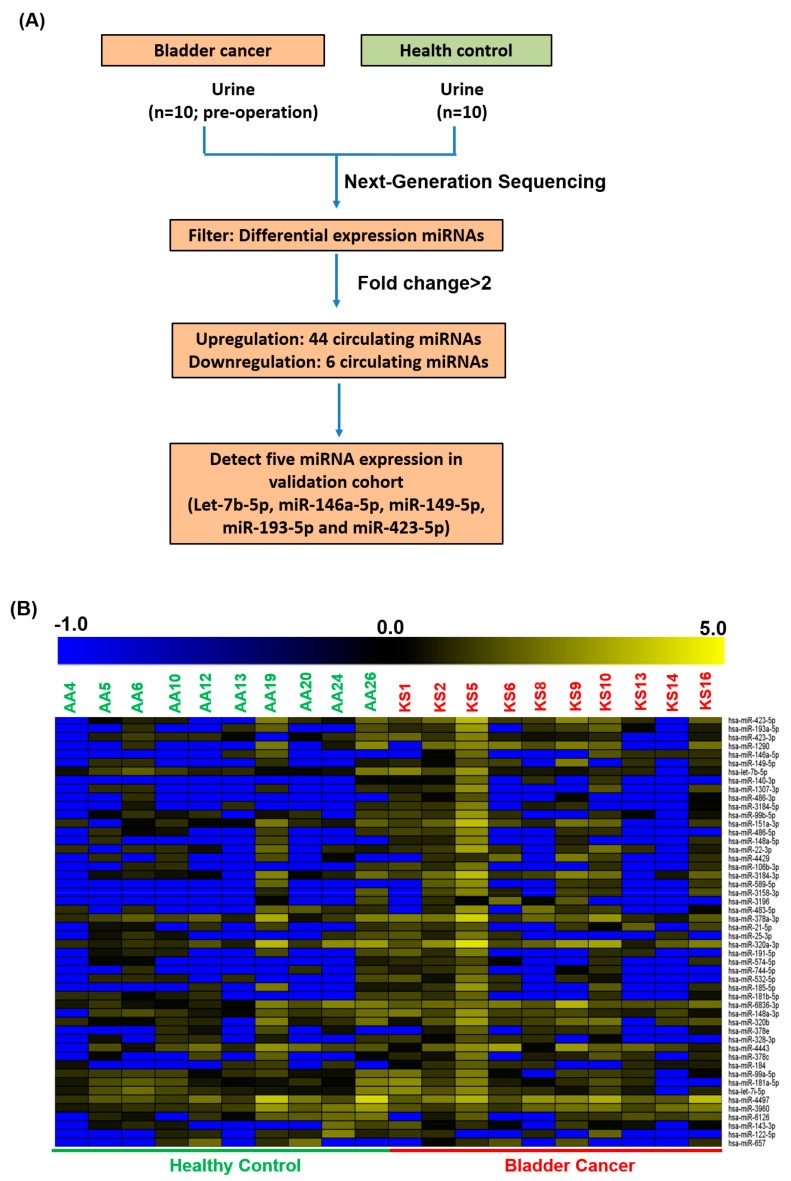
Differentially expressed circulating miRNAs were identified in urine from healthy volunteers and patients with bladder cancer using next-generation sequencing. (**A**) Flowchart of 10 NGS profiles. Circulating miRNAs with differential expression were filtered with a fold change of ≥2 or <0.25. (**B**) Heatmap presenting the upregulation or downregulation of circulating miRNAs in patients with bladder cancer. (**C**) Targets of differentially expressed miRNA candidates were identified using a target prediction tool and pathway enrichment analysis.

**Figure 2 ijms-22-04278-f002:**
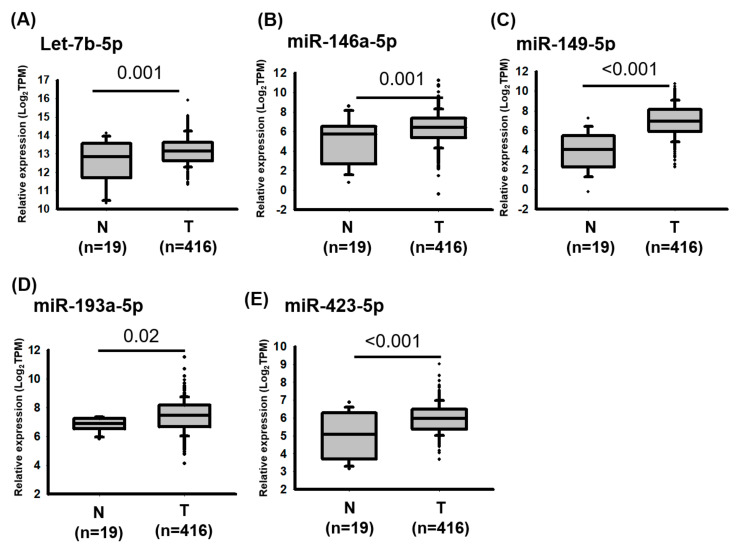
Expression levels of miRNA candidates in bladder cancer were revealed through an analysis of the TCGA database. Expression levels of (**A**) *let-7b-5p*, (**B**) miR-146-5p, (**C**) *miR-149-5p*, (**D**) *miR-193a-5p* and (**E**) *miR-423-5p* were examined by analyzing The Cancer Genome Atlas database. The transcriptome profiles of 435 patients with bladder cancer were downloaded from said database, including 19 adjacent normal tissues and 416 bladder cancer tissues. The expression levels of miRNAs are presented in transcripts per million (TPM). Those of individual miRNAs were analyzed using Student’s *t* tests.

**Figure 3 ijms-22-04278-f003:**
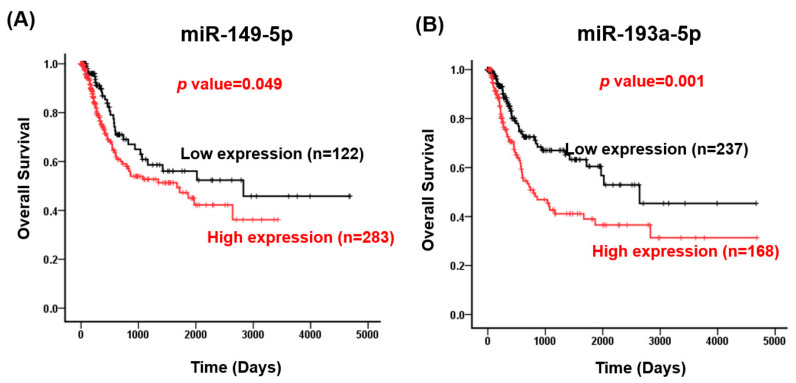
Expression levels of *miR-149-5p* and *miR-193a-5p* were highly associated with the survival curve of patients with bladder cancer. The receiver operating characteristic curve was used to define cutoff values for *miR-149-5p* and *miR-193-5p* expression levels. According to the defined cutoff values (69 for *miR-149-5p* and 199 for *miR-193-5p*), the patients were divided into groups according to whether they had high or low expression levels of *miR-149-5p* or *miR-193-5p*. The correlation between *miR-149-5p* expression (**A**) and *miR-193a-5p* expression (**B**) was analyzed according to the overall survival curve.

**Figure 4 ijms-22-04278-f004:**
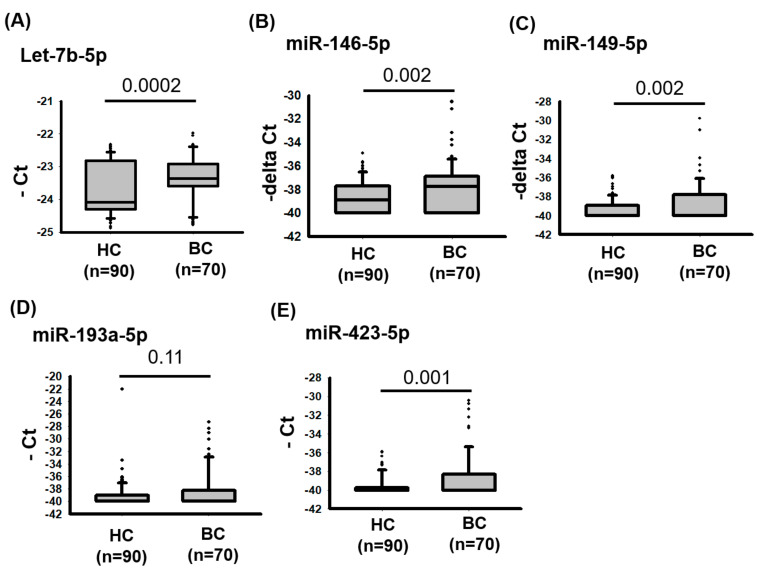
Expression levels of *let-7b-5p*, *miR-146a-5p*, *miR-149-5p*, *miR-193a-5p*, and *miR-423-5p* in urine from healthy controls and patients with bladder cancer. Relative levels of (**A**) *let-7b-5p*, (**B**) *miR-146a-5p*, (**C**) *miR-149-5p*, (**D**) *miR-193-5p*, and (**E**) *miR-423-5p* were examined in urine from 90 healthy controls and 70 patients with bladder cancer using TaqMan real-time PCR Assays.

**Table 1 ijms-22-04278-t001:** Univariate and multivariate Cox’s regression analysis of gene expression for overall survival of 405 patients with bladder cancer.

Characteristic	No. (%)	OS			
		CHR (95% CI)	*p*-Value	AHR (95% CI)	*p*-Value
***Let-7b-5p***					
Low	151 (37.3)	1.00		1.00	
High	254 (62.7)	1.39 (0.92–2.10)	0.120	1.34 (0.89–2.03)	0.167
***miR-149-5p***					
Low	122 (30.1)	1.00		1.00	
High	283 (69.9)	1.52 (1.00–2.33)	0.050	1.70 (1.11–2.61)	0.015
***miR-146a-5p***					
Low	68 (16.8)	1.00		1.00	
High	337 (83.2)	1.12 (0.64–1.97)	0.688	0.94 (0.53–1.66)	0.832
***miR-193a-5p***					
Low	237 (58.5)	1.00		1.00	
High	168 (41.5)	1.90 (1.29–2.78)	0.001	1.86 (1.26–2.74)	0.002
***miR-423-5p***					
Low	30 (7.4)	1.00		1.00	
High	375 (92.6)	1.84 (0.68–4.99)	0.233	2.09 (0.77–5.70)	0.148

Abbreviation: OS, overall survival; CHR, crude hazard ratio; AHR, adjusted hazard ratio. AHR were adjusted for AJCC pathological stage (II, III and IV VS. I).

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
