# Peer review of "Circulating miRNAs Act as Diagnostic Biomarkers for Bladder Cancer in Urine"

_ijms, 2021, doi:10.3390/ijms22084278_

Round 1
Reviewer 1 Report
The authors present a study aiming to identify circular miRNA that can be used as a diagnostic tool in bladder cancer. After the first selection of 50 miRNA from a next-generation sequencing analysis of 20 individuals (10 cancer, 10 controls), the authors selected 4 miRNAs and they validated them in a large cohort of patients. I found the paper well-written and with an appropriate design. I have only minor comments that a list here below:
1-The characters of Figure 1B and 1C are too small
2- Please, select another color code for Figure 1B to meet the needs of colorblind readers
3- Improve the enrichment analysis using different tools, such as MIENTURNET. I suggest including an analysis of the genes targeted by the miRNA selected
4- The authors could use a logarithm scale in Figure 2.
Author Response
Dear Editor~
We would like to resubmit the enclosed manuscript “Circulating miRNAs Act as Diagnostic Biomarkers for Bladder Cancer in Urine” by Jen-Tai Lin et al. for consideration for publication in International Journal of Molecular Sciences. Based on the reviewers’ comments, we further modified the Figure 1B, 1C and Figure 2. Furthermore, we added a paragraph to describe previous research on urine-related biomarkers for bladder cancer. Changes to the manuscript are highlighted in yellow. We had replied to the reviewers’ questions point-by-point in the response letter.
Reviewer #1
The authors present a study aiming to identify circular miRNA that can be used as a diagnostic tool in bladder cancer. After the first selection of 50 miRNA from a next-generation sequencing analysis of 20 individuals (10 cancer, 10 controls), the authors selected 4 miRNAs and they validated them in a large cohort of patients. I found the paper well-written and with an appropriate design. I have only minor comments that a list here below:
1. The characters of Figure 1B and 1C are too small
Response: We have added a new figure with high resolution to the revised version of the manuscript.
2. Please, select another color code for Figure 1B to meet the needs of colorblind readers
Response: We thank the reviewer for bringing this to our attention. We have replaced red and green with yellow and blue, respectively in Figure 1B.
3. Improve the enrichment analysis using different tools, such as MIENTURNET. I suggest including an analysis of the genes targeted by the miRNA selected
Response: In accordance with the reviewer’s suggestion, we conducted a pathway enrichment analysis using MIENTURNET. The results were highly similar to our original results. We have added new data (Figure 1C) to the revised version of the manuscript as well as a description of this new data to the Results section. The Methods section has also been modified.
The following text has been added to the Results section, page 6:
……..expressed miRNAs and subjected them to pathway enrichment analysis. 【As illustrated in Figure 1C, the biological functions of these miRNA candidates were significantly involved in cancer-related signaling pathways, including those of proteoglycans in cancer, MAPK, TGF-beta, FoxO, colorectal cancer, cellular senescence, the adherens junction, PI3K-Akt, Hippo, autophagy, focal adhesion, and ErbB. 】Some of these signaling pathways have been reported to have a biological function in bladder cancer progression[21,22].
4- The authors could use a logarithm scale in Figure 2.
Response: In accordance with the reviewer’s suggestion, we attempted to use a logarithm scale (log2) to present our data. Because the expression levels of miR-1246 and miR-1290 were very low, we omit these results in the revised version of the manuscript. We have also modified our manuscript accordingly.
In results section, page 7:
…………..Therefore, we selected five upregulated circulating miRNAs, namely let-7b-5p, miR-146-5p, miR-149-5p, miR-423-5p and miR-193a-5p, miR-1290, and miR-1246, for further examination in bladder cancer……….
Reviewer#2
1. In figure 3, authors should explain how each case was divided into low or high expression.
Response: We apologize for not providing a clear description. We downloaded all the transcriptome data and clinical information of 416 patients with bladder cancer from the TCGA database. After the exclusion of patients for whom survival information was lacking, 405 patients were included in the survival analysis (Figure 3). We reanalyzed the effects of miR-193-5p and miR-149-5p expression on the survival curve of bladder cancer by using the data of the aforementioned 405 patients from the TCGA database. We have added the optimal cutoff values to the legend of Figure 3 in the revised version of the manuscript.
In page 8;
Figure 3. Expression levels of miR-149-5p and miR-193a-5p were highly associated with the survival curve of patients with bladder cancer. 【The receiver operating characteristic curve was used to define cutoff values for miR-149-5p and miR-193-5p expression levels. According to the defined cutoff values (69 for miR-149-5p and 199 for miR-193-5p), the patients were divided into groups according to whether they had high or low expression levels of miR-149-5p or miR-193-5p. The correlation between miR-149-5p expression (A) and miR-193a-5p expression (B) was analyzed according to the overall survival curve.】
2. Was the expression of the miRNA in figure 2 or 4 associated with tumor volume of the bladder cancer?
Response: We further analyzed the clinical impact of miRNA candidate expression on patients with bladder cancer by using data from the TCGA database. Our data revealed that miRNA expression levels were not strongly associated with tumor volume. Because of the lack of significance of these data, we do not present them in the revised version of the manuscript.
3. Are there other urine biomarkers such as protein, DNA and other RNA? Authors should describe the utility of urine miRNA as a biomarker compared with other molecules.
Response: We thank the reviewer for this suggestion. We have added further details regarding urine biomarkers to the Introduction section.
The following text has been added to the Introduction section, page 3:
This difficulty has led scientists to develop new cystoscopic techniques, such as narrow-band imaging cystoscopy and photodynamic diagnosis. 【Previous studies have identified numerous noninvasive biomarkers in urine that can serve as diagnostic indicators of bladder cancer [3]. Comprehensive screening approaches have been applied to urine samples to identify cell-free-DNA, cell-free-RNA, small RNA, or DNA methylation in cells, including DNA methylation of PCDH17 and POU4F2 [4], as well as determine circulating mRNA levels of IGFBP5, HOXA13, MDK, CDK1, and CXCR2 [5] and the mutation status of FGFR3 and TERT promoters [6]. Some urine-related biomarkers have been approved for clinical use by the United States Food and Drug Administration (FDA), such as nuclear matrix protein 22 [7], human complement factor H-related protein [8], carcinoembryonic antigen [9], and sulfated mucin glycoprotein [10]. These biomarkers are typically used in conjunction with cystoscopy to increase the diagnostic sensitivity and specificity for bladder cancer. However, these FDA-approved biomarkers have the shortcoming of a high false positive rate in patients with inflammatory conditions.】To date, the diagnosis of early bladder cancer remains difficult; therefore, the development of noninvasive biomarkers for early diagnosis will be highly beneficial for patients with bladder cancer.

Reviewer 2 Report
In figure 3, authors should explain how each case was divided into low or high expression.
Was the expression of the miRNA in figure 2 or 4 associated with tumor volume of the bladder cancer?
Are there other urine biomarkers such as protein, DNA and other RNA? Authors should describe the utility of urine miRNA as a biomarker compared with other molecules.
Author Response
Dear Editor~
We would like to resubmit the enclosed manuscript “Circulating miRNAs Act as Diagnostic Biomarkers for Bladder Cancer in Urine” by Jen-Tai Lin et al. for consideration for publication in International Journal of Molecular Sciences. Based on the reviewers’ comments, we further modified the Figure 1B, 1C and Figure 2. Furthermore, we added a paragraph to describe previous research on urine-related biomarkers for bladder cancer. Changes to the manuscript are highlighted in yellow. We had replied to the reviewers’ questions point-by-point in the response letter.
Reviewer #1
The authors present a study aiming to identify circular miRNA that can be used as a diagnostic tool in bladder cancer. After the first selection of 50 miRNA from a next-generation sequencing analysis of 20 individuals (10 cancer, 10 controls), the authors selected 4 miRNAs and they validated them in a large cohort of patients. I found the paper well-written and with an appropriate design. I have only minor comments that a list here below:
1. The characters of Figure 1B and 1C are too small
Response: We have added a new figure with high resolution to the revised version of the manuscript.
2. Please, select another color code for Figure 1B to meet the needs of colorblind readers
Response: We thank the reviewer for bringing this to our attention. We have replaced red and green with yellow and blue, respectively in Figure 1B.
3. Improve the enrichment analysis using different tools, such as MIENTURNET. I suggest including an analysis of the genes targeted by the miRNA selected
Response: In accordance with the reviewer’s suggestion, we conducted a pathway enrichment analysis using MIENTURNET. The results were highly similar to our original results. We have added new data (Figure 1C) to the revised version of the manuscript as well as a description of this new data to the Results section. The Methods section has also been modified.
The following text has been added to the Results section, page 6:
……..expressed miRNAs and subjected them to pathway enrichment analysis. 【As illustrated in Figure 1C, the biological functions of these miRNA candidates were significantly involved in cancer-related signaling pathways, including those of proteoglycans in cancer, MAPK, TGF-beta, FoxO, colorectal cancer, cellular senescence, the adherens junction, PI3K-Akt, Hippo, autophagy, focal adhesion, and ErbB. 】Some of these signaling pathways have been reported to have a biological function in bladder cancer progression[21,22].
4. The authors could use a logarithm scale in Figure 2.
Response: In accordance with the reviewer’s suggestion, we attempted to use a logarithm scale (log2) to present our data. Because the expression levels of miR-1246 and miR-1290 were very low, we omit these results in the revised version of the manuscript. We have also modified our manuscript accordingly.
In results section, page 7:
…………..Therefore, we selected five upregulated circulating miRNAs, namely let-7b-5p, miR-146-5p, miR-149-5p, miR-423-5p and miR-193a-5p, miR-1290, and miR-1246, for further examination in bladder cancer……….
Reviewer#2
1. In figure 3, authors should explain how each case was divided into low or high expression.
Response: We apologize for not providing a clear description. We downloaded all the transcriptome data and clinical information of 416 patients with bladder cancer from the TCGA database. After the exclusion of patients for whom survival information was lacking, 405 patients were included in the survival analysis (Figure 3). We reanalyzed the effects of miR-193-5p and miR-149-5p expression on the survival curve of bladder cancer by using the data of the aforementioned 405 patients from the TCGA database. We have added the optimal cutoff values to the legend of Figure 3 in the revised version of the manuscript.
In page 8;
Figure 3. Expression levels of miR-149-5p and miR-193a-5p were highly associated with the survival curve of patients with bladder cancer. 【The receiver operating characteristic curve was used to define cutoff values for miR-149-5p and miR-193-5p expression levels. According to the defined cutoff values (69 for miR-149-5p and 199 for miR-193-5p), the patients were divided into groups according to whether they had high or low expression levels of miR-149-5p or miR-193-5p. The correlation between miR-149-5p expression (A) and miR-193a-5p expression (B) was analyzed according to the overall survival curve.】
2. Was the expression of the miRNA in figure 2 or 4 associated with tumor volume of the bladder cancer?
Response: We further analyzed the clinical impact of miRNA candidate expression on patients with bladder cancer by using data from the TCGA database. Our data revealed that miRNA expression levels were not strongly associated with tumor volume. Because of the lack of significance of these data, we do not present them in the revised version of the manuscript.
3. Are there other urine biomarkers such as protein, DNA and other RNA? Authors should describe the utility of urine miRNA as a biomarker compared with other molecules.
Response: We thank the reviewer for this suggestion. We have added further details regarding urine biomarkers to the Introduction section.
The following text has been added to the Introduction section, page 3:
This difficulty has led scientists to develop new cystoscopic techniques, such as narrow-band imaging cystoscopy and photodynamic diagnosis. 【Previous studies have identified numerous noninvasive biomarkers in urine that can serve as diagnostic indicators of bladder cancer [3]. Comprehensive screening approaches have been applied to urine samples to identify cell-free-DNA, cell-free-RNA, small RNA, or DNA methylation in cells, including DNA methylation of PCDH17 and POU4F2 [4], as well as determine circulating mRNA levels of IGFBP5, HOXA13, MDK, CDK1, and CXCR2 [5] and the mutation status of FGFR3 and TERT promoters [6]. Some urine-related biomarkers have been approved for clinical use by the United States Food and Drug Administration (FDA), such as nuclear matrix protein 22 [7], human complement factor H-related protein [8], carcinoembryonic antigen [9], and sulfated mucin glycoprotein [10]. These biomarkers are typically used in conjunction with cystoscopy to increase the diagnostic sensitivity and specificity for bladder cancer. However, these FDA-approved biomarkers have the shortcoming of a high false positive rate in patients with inflammatory conditions.】To date, the diagnosis of early bladder cancer remains difficult; therefore, the development of noninvasive biomarkers for early diagnosis will be highly beneficial for patients with bladder cancer.

Round 2
Reviewer 2 Report
I have no more comments.